# AI cancer driver mutation predictions are valid in real-world data

Thinh N. Tran ⓘ, Chris Fong, Karl Pichotta, Anisha Luthra, Ronglai Shen ⓘ, Yuan Chen ⓘ, Michele Waters ⓘ, Susie Kim, Xiang Li ⓘ, Ino de Bruijn ⓘ, Gregory Riely, Michael F. Berger ⓘ, Marc Ladanyi ⓘ, Debyani Chakravarty ⓘ, Nikolaus Schultz ⓘ ✉ & Justin Jee ⓘ ✉

Characterizing and validating which mutations influence development of cancer is challenging. Artificial intelligence (AI) has delivered significant advances in protein structure prediction, but its utility for identifying cancer drivers is less explored. We evaluate multiple computational methods for identifying cancer driver mutations. For re-identifying known drivers, methods incorporating protein structure or functional genomic data outperform methods trained only on evolutionary data. We validate variants of unknown significance (VUSs) annotated as pathogenic by testing their association with overall survival in two cohorts of patients with non-small cell lung cancer (N = 7965 and 977). VUSs identified as pathogenic drivers by AI in *KEAP1* and *SMARCA4* are associated with worse survival, unlike "benign" VUSs. "Pathogenic" VUSs also exhibit mutual exclusivity with known oncogenic alterations at the pathway level, further suggesting biological validity. AI predictions thus contribute to a more comprehensive understanding of tumor genetics as validated by real-world data.

The majority of somatic tumor mutations are variants of unknown significance (VUSs)[1–3]. In a pan-cancer, multi-institutional cohort of N = 160,969 patients with tumor genomic profiling[4], approximately 80% of somatic mutations detected were VUSs according to an FDA-recognized molecular knowledge database (OncoKB[5]). In some genes with known consequences for survival, such as *KEAP1*[6], 78.8% were VUSs (Fig. 1A).

Multiple knowledge bases have been developed to annotate pathogenic and actionable mutations[5,7,8], however, these generally rely on published literature, which is time-consuming to produce and to compile. Computational variant effect predictors (VEPs) may automate variant annotation. Recent tools such as Google DeepMind's Alpha-Missense, which leverage evolutionary, biological and protein structural features combined with high-dimensional machine learning architectures, have gained significant interest[9,10]. However, these VEPs are generally trained to predict germline pathogenic variants, and their utility in identifying somatic mutations that drive diseases such as cancer remains uncharacterized[9,11–15]. The validation of such tools is

itself a challenging task; functional assays are labor-intensive and can thus characterize only a limited number of variants[3,16,17].

In this work, we develop four specific approaches to assess the utility of computational methods for annotating VUSs using real-world patient cancer data. Digital health records and widespread tumor genomic profiling offer a means to bypass traditional functional assays and study the impact of tumor mutations, including VUSs, directly using patient data[18,19]. Our four approaches include 1) annotating known pathogenic somatic cancer variants as confirmed by OncoKB, which combines literature-confirmed annotations with population-level hotspot identification[20], 2) identifying VUSs associated with binding regions among proteins with known structures, 3) identifying VUSs associated with overall survival (OS) in patients with lung cancer and 4) identifying VUSs in tumors without other drivers in the same oncogenic pathway. We apply these four approaches to evaluate 14 modern computational methods chosen based on their conceptual advancements and demonstrated superior performance in annotating known pathogenic mutations in databases such as ClinVar[21] and

Memorial Sloan Kettering Cancer Center, New York, NY, USA. ✉e-mail: schultzn@mskcc.org; jeej@mskcc.org

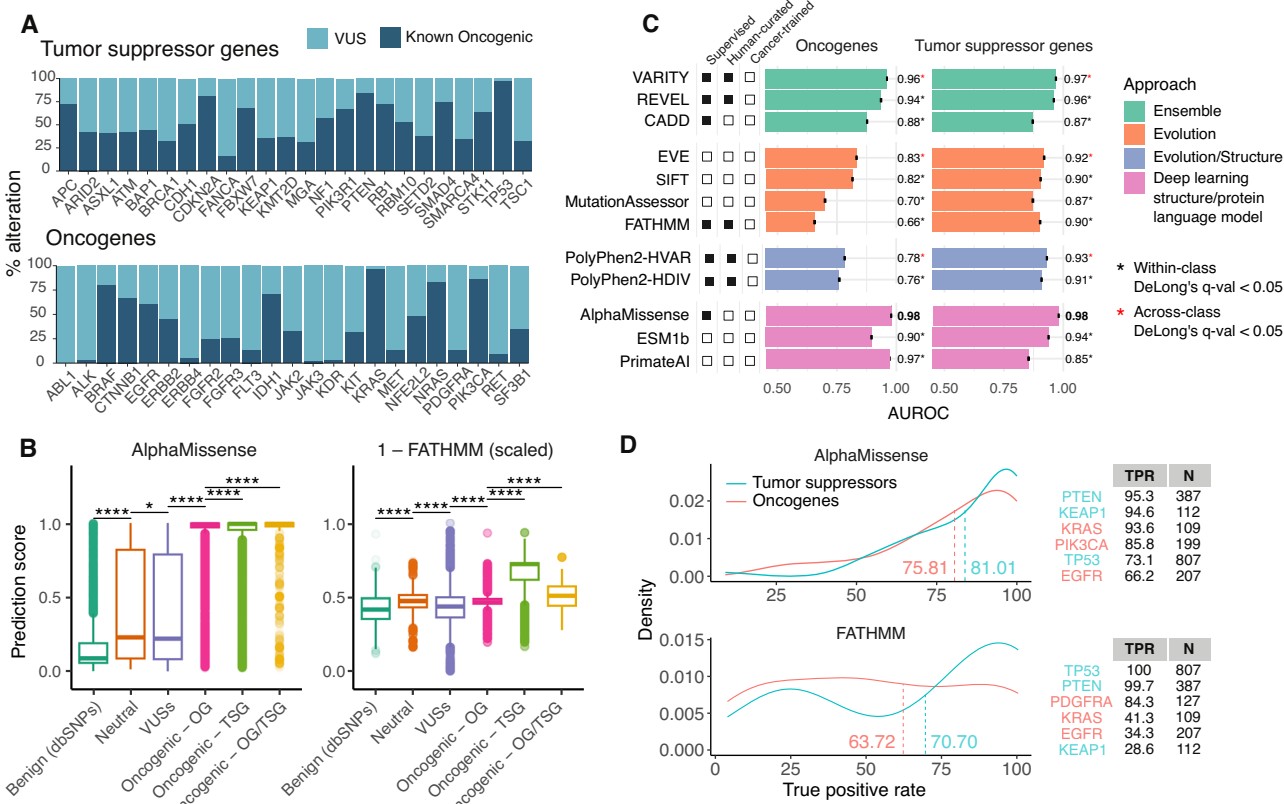

**Fig. 1 | VEPs have variable performance in annotating known oncogenic mutations.** **A** Frequency of known oncogenic mutations and variants of unknown significance (VUS) in commonly altered oncogenes and tumor suppressor genes in GENIE as annotated by OncoKB. **B** Distributions of prediction scores from Alpha-Missense and FATHMM from non-pathogenic dbSNPs ($N = 7474$) and missense mutations in GENIE v.14-public, broken down by their occurrence in oncogenes (OG, $N = 408,771$), tumor suppressor genes (TSG, $N = 506,068$) or genes that act as both (OG/TSG, $N = 57,592$) at the population level, in which all occurrences of missense mutations are included. Points higher on the y-axis corresponded with higher predicted pathogenicity. Boxplots represent mean scores (center line) ± interquartile range (IQR); whiskers span 1.5 × IQR from each quartile, with outliers shown individually. See Fig. S3B for population-level distributions from all VEPs and Fig. S3A for mutation-level distributions. Brackets denote significance from two-sided Tukey's tests with FDR correction (*: $q \le 0.05$, ****: $q \le 1e\text{-}04$). **C.** AUROCs (± 95%CI) of 12 variant annotation methods in classifying known oncogenic mutations

($N = 180,540$) and non-oncogenic SNPs ($N = 180,540$ upsampled from 7474) at the population level. DeLong's test was used to compare AUROCs with FDR correction. Within each methodological class, pairwise comparisons were performed between the top-performing method and others (*: $q \le 0.05$ marked by black asterisks). Red asterisks denote significant differences ($q \le 0.05$) between each class's top performer and the overall best method (bolded AUROC). Tracks at left indicate how each method was trained: "Supervised" denotes use of labeled training data; "Human-curated" specifies whether labels originated from manually curated resources (e.g., ClinVar); and "Cancer-trained" denotes use of cancer-specific datasets (e.g., Cancer Genome Census). **D.** Density plots showing true positive rates (TPR) of AlphaMissense and FATHMM over all genes. TPRs and the number of known oncogenic mutations (N) in select commonly mutated oncogenes and tumor suppressor genes are shown. See Supplemental Appendix for a complete list of TPRs. Source data are provided as a Source Data file.

VariBench[22] compared to other methods in the same class[23,24] (Table S1). Our results show that methods incorporating structural or functional genomic features outperform those relying solely on evolutionary conservation when identifying known cancer drivers. VUSs predicted to be pathogenic—particularly in genes like *KEAP1* and *SMARCA4*—are consistently associated with poorer overall survival in two nonoverlapping non-small cell lung cancer cohorts, and tend to be mutually exclusive with other known oncogenic alterations within the same pathways. These results reinforce the clinical and biological relevance of the predictions, highlighting the value of computational VEPs in advancing the interpretation of somatic variants in cancer.

## Results

### Association with OncoKB driver variants

We first tested the utility of VEPs in discriminating literature-confirmed or hotspot pathogenic somatic missense variants from benign ones. OncoKB-annotated pathogenic variants in the AACR Project GENIE dataset[4] served as pathogenic cases while randomly selected missense mutations from the dbSNP Human Variation Sets labeled as having no

known medical impact served as negative controls. Since germline missense variants from dbSNPs may not fully capture the complexity of somatic mutational processes, particularly given the differences in mutagenesis between germline and somatic tissues, we created an additional negative control set including simulated neutral variants using tri-nucleotide change probabilities observed in the mutational profiles of the specific tumor types in GENIE v14 (simulated SNPs). Pathogenicity predictions from all methods were generally correlated with each other at both the mutation level, where each unique mutation was counted once, and at the population level, where all occurrences of mutations were counted to reflect actual population frequencies of each mutation (Fig. S2).

Benign SNPs had significantly lower scores than oncogenic mutations in all studied methods, demonstrating their ability to distinguish benign versus pathogenic mutations in cancer (Figs. 1B and S3, SA: Predicting Established Pathogenic Variants). Mutations in GENIE annotated as neutral or unknown by OncoKB had significantly higher scores than known benign mutations, suggesting potential unannotated drivers (Fig. 1B). Across all methods, oncogenic

mutations in tumor suppressor genes (TSGs) had higher predicted scores and higher AUROC for correctly annotating oncogenic mutations in TSGs than in oncogenes (OGs) (Fig. 1C), as expected based on previous work[25]. These results were consistent when simulated SNPs were used as the negative class (Figs. S12, S13).

We found that in general, the ensemble and deep learning-based methods outperformed the evolution-based methods (Fig. 1C). AlphaMissense[9] significantly outperformed other deep learning-based methods as well as other best-in-class methods in predicting oncogenic mutations (AUROC of 0.98 for OGs and TSGs at the population level respectively, Fig. 1C). Among ensemble methods, VARITY[26] and REVEL[14], both trained on human-curated data, outperformed CADD[27], which was trained on weak population-derived labels (Fig. 1C). Among evolution-based methods, EVE[28], the only unsupervised deep learning method in this class, outperformed others at the population level (AUROC of 0.83 and 0.92 for OGs and TSGs respectively, Fig. 1C). These results held when mutant alleles were each counted once, irrespective of population frequency (Fig. S4), and when using simulated SNPs as the negative class (Fig. S13). Additionally, to more accurately reflect the type and distribution of passenger mutations in tumors, we used neutral somatic mutations as annotated by OncoKB as the negative class to evaluate VEP performance (Fig. S17). We found that performance declined across methods, even though deep learning and ensemble methods tended to perform better than evolution-based methods (Fig. S17). Out of all methods, AlphaMissense achieved the best performance in this task (AUROC of 0.8 for classifying OGs and TSGs mutations at the population level, respectively, Fig. S17B). Beyond general VEPs, we evaluated two methods that leverage tumor type-specific information, such as recurrence and mutations clustering in 3D structure, to train their predictive models: CHASMplus[29] and BoostDM[30], the latter of which includes predictions for a smaller number of mutations (Figure S16). Both performed well in identifying oncogenic mutations at the population level, although BoostDM's performance was lower at the mutation level (Fig. S16B), possibly because it focused on a small number of very common mutations in cancer. Across all methods, sensitivity was higher in TSGs compared to OGs, but sensitivity further varies at the gene level (Fig. 1D). Overall, these results demonstrate that VEPs were able to identify pathogenic mutations in cancer, with multimodal, deep learning-based methods outperforming methods trained only on mutation frequencies.

It is possible that VEPs with different approaches may have complementary information that could result in better performance in predicting variant effects than any single VEP. To test this hypothesis, we trained random forest (RF) models including the outputs from 11 non-ensemble VEPs as inputs to predict variant pathogenicity. Our training and test sets included OncoKB oncogenic variants from GENIE v14 data as the positive class and randomly selected variants from dbSNP as the negative class. Models trained and tested on population data perform better than those trained and tested on mutation level data; 5-fold CV models perform better than gene holdout models; and all models perform better in predicting TSG mutations compared to OGs (Fig. S15). The best performing ensemble, trained on population-level data and validated using 5-fold cross-validation, achieved AUCs of 0.998 on predicting both TSG and OG mutations, outperforming the best performing VEP Alpha-Missense (DeLong's Test: $p = 2.6e-51$, $\Delta$AUC = 0.034, 95% CI = [0.029, 0.038] in TSG, $p = 4.4e-75$, $\Delta$AUC = 0.055, 95% CI = [0.049, 0.061] in OG, Fig. S15C). Feature importance scores from all ensembles consistently identify AlphaMissense, CHASMplus and PrimateAI as the top three most important features. These results suggest that ensemble predictors at the population level were able to incorporate knowledge from individual methods, particularly from well-performing VEPs, in their predictions, which resulted in improved performance over non-ensemble methods.

## Association with known binding sites

We next investigated the ability of VEPs to identify new driver mutations not previously annotated by OncoKB from the large pool of detected VUSs. In particular, we validated the potential functional impact of VUSs labeled as pathogenic by VEPs ("reclassified pathogenic") in cancer through analyses of their impact on protein binding sites, correlation with patient outcomes, and adherence to expected driver mutual exclusivity patterns. Furthermore, we implemented success metrics on each task and compared the performance of VEPs in order to identify the best method for studying new driver mutations.

Pathogenic mutations can alter protein function by disrupting interactions with other proteins and ligands[31]. We probed whether reclassified pathogenic variants were enriched in residues involved in ligand binding or protein-protein interaction ("binding residues") for proteins with available crystal structures. Mutations affecting binding residues in all genes were significantly more likely to be annotated as oncogenic by OncoKB as expected (Fisher's test, $q$-value = 0, odds ratio = 10.02, 95% CI = [9.45, 10.63], Fig. 2A). Mutations occurring at binding residues were universally more likely to be reclassified as pathogenic, whereas non-binding residue mutations were more likely to be reclassified as benign, although the degree of enrichment varied by method (Fig. 2A). This result suggests that the disruption of function at these critical binding residues may contribute to the pathogenic nature of these reclassified pathogenic variants. To quantify performance of VEPs in reclassifying VUSs at binding residues to be pathogenic by calculating the odds ratio of reclassified pathogenic mutations occurring in a binding domain compared to reclassified benign mutations across all genes ($OR_{binding}$) (Fig. S14B). VARITY achieves the highest $OR_{binding}$ of 9.09 (95% CI [8.68, 9.53]) followed by Alpha-Missense ($OR_{binding}$ of 7.1, 95% CI [6.81, 7.41]), meaning reclassified pathogenic mutations identified by these VEPs are ~7-9 times more likely to occur at binding sites compared to reclassified benign mutations (Fig. S14B). This suggests that VARITY and AlphaMissense perform best in distinguishing VUSs according to physical location within the protein.

## Association with survival

To further validate VUS classifications by VEPs using real-world data, we measured the impact of mutations according to classification on overall survival (OS), focusing on patients with non-small cell lung cancer (NSCLC), the world's leading cause of cancer mortality[32] and a cancer type with frequent tumor genomic sequencing. In patients with NSCLC, mutations in multiple genes, including *KRAS*, *STK11* and *KEAP1*, have been associated with worse OS[33–36]. We investigated whether reclassified pathogenic variants are associated with OS in NSCLC using two cohorts of patients: 7,965 patients with MSK-IMPACT clinical sequencing and 977 non-MSK patients from the GENIE Biopharma Collaborative (BPC) NSCLC cohort[37]. To identify the association between reclassified pathogenic variants and outcome, we stratified patients based on gene-level pathogenicity annotations and compared OS between groups using Cox's proportional hazard (PH) models. To account for potential covariate imbalances between groups with distinct genomic profiles, we calculated inverse probability of treatment weights (IPTW) using demographic, genomic and clinical covariates prior to fitting Cox PH models, which demonstrates effectiveness in address these imbalances, especially in confounding genomic variables such as tumor mutational burden (SA: Survival Analysis, Figure S5).

Known oncogenic variants[3] in several genes were associated with worse OS. VUSs in *KEAP1* and *SMARCA4* annotated pathogenic by multiple methods were also associated with worse OS, while those annotated as likely benign were associated with better outcome, suggesting meaningful discrimination among VUSs by computational methods (Fig. 2B). These findings were consistent in both the MSK-

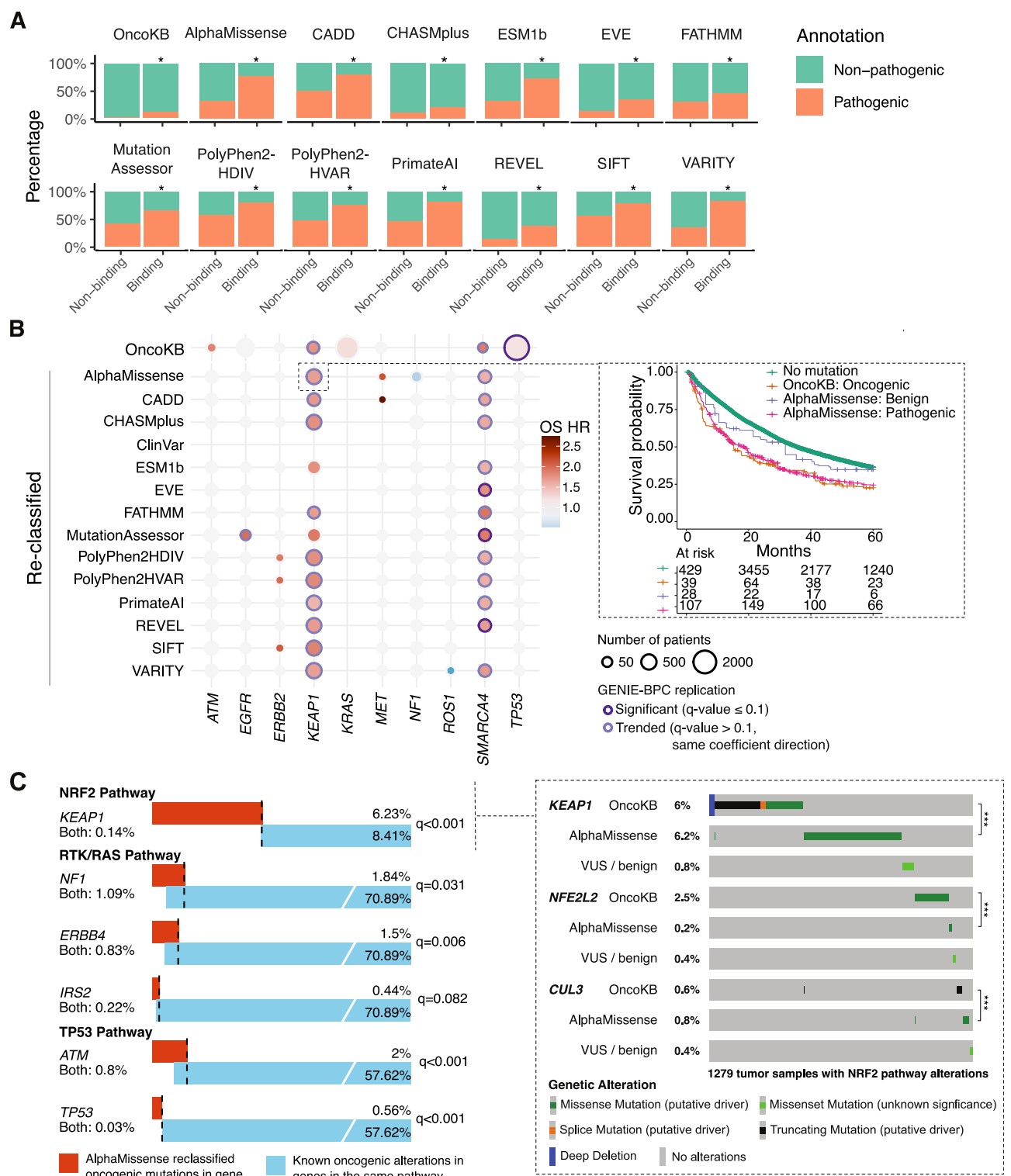

**Fig. 2 | VEPs identify unannotated driver mutations. A** Frequency and annotation of missense mutations occurring at binding residues (either ligand binding or protein-protein interaction hotspots, see Supplemental Appendix) or non-binding residues of all genes with available binding residue information in GENIE v14-public pan-cancer cohort (*N* = 209,588). Asterisks denote statistical significance from two-sided Fisher's exact tests with FDR correction (*: *q*-value ≤ 0.1). OncoKB groups include all missense mutations, whereas variant effect predictor groups only include VUSs. **B** Inverse probability treatment weighted overall survival hazard ratios (from time of diagnosis left truncated at time of sequencing) of patients harboring reclassified oncogenic mutations compared to patients without mutation in commonly mutated genes in non-small cell lung cancer (NSCLC). Patients are from MSK-IMPACT NSCLC (*N* = 7965) and AACR GENIE Biopharma

Collaborative NSCLC (*N* = 977) cohorts. Inset: Inverse probability of treatment weighted Kaplan Meier curves comparing overall survival from time of diagnosis left-truncated at time of sequencing of patients based on *KEAP1* mutations annotation in the MSK-IMPACT NSCLC cohort. Alteration frequencies and overlap of AlphaMissense reclassified pathogenic mutations with oncogenic alterations of genes in the same pathway in MSK-IMPACT NSCLC cohort (*N* = 7965 patients). Inset: Oncoprint of genes in the NRF2 pathway for *N* = 1279 samples with NRF2 pathway alteration. Asterisks denote statistical significance from two-sided Fisher's exact tests with FDR correction (***: *q*-value ≤ 0.01). *KEAP1* reclassified mutations, similar to *KEAP1* oncogenic mutations, are mutually exclusive with other oncogenic mutations in *NFE2L2* and *CUL3*. Source data are provided as a Source Data file.

IMPACT and BPC cohorts (Fig. 2B). The higher OS risks in patients with reclassified pathogenic mutations in these two genes were comparable with the risks associated with known oncogenic mutations, suggesting high specificity for pathogenic variant detection across methods (Fig. S6A). Conversely, patients with reclassified benign mutations in these genes had some differences in OS risk compared to those with no mutation; for example, CADD, FATHMM, and PrimateAI appeared to have imperfect sensitivity for pathogenic *KEAP1* mutations (Fig. S6B). We quantified VEP performance in distinguishing VUSs with outcome implications by calculating the relative risk of survival (RR), defined for each gene as the ratio of OS HR for patients with reclassified oncogenic mutations vs. no mutation, compared to OS HR for patients with reclassified benign mutations vs. no mutation[38] (Fig. S14C). The higher the RR, the better a VEP at identifying mutations that affect outcomes. The RR for *KEAP1* ($RR_{KEAP1}$) showed that SIFT and PolyPhen2-HDIV achieved the best performance ($RR_{KEAP1}$ range 1.57-1.7, 95% CI [1.47, 1.81]), followed by MutationAssessor, AlphaMissense and PolyPhen2-HVAR at ($RR_{KEAP1}$ range 1.32-1.43, 95% CI [1.24, 1.52]) (Fig. S14C). This result suggests that evolution-based methods successfully identified prognostic mutations, even when they performed less well in annotating known oncogenic mutations.

Concurrent mutations in certain gene combinations may worsen survival in an additive manner, as is known for *STK11* and *KEAP1* in NSCLC[34] and observed in the MSK-IMPACT NSCLC cohort (Fig. S9B). To test whether reclassified pathogenic variants in *STK11* and *KEAP1* were similarly associated with worse survival, we compared OS of patients with double *KEAP1* and *STK11* reclassified mutations with patients with single reclassified pathogenic mutation and patients without any mutation in these two genes. We found that patients with tumors harboring both *KEAP1* and *STK11* AlphaMissense mutations had worse OS compared to those with reclassified pathogenic mutations in either genes, as well as those without mutation (Fig. S9A). This result further suggests that the *KEAP1* and *STK11* reclassified variants from AlphaMissense follow expected patterns of additive prognostic significance, suggesting biologic validity. In summary, VEP annotations suggested several VUSs with biological activity that were confirmed by association with OS in NSCLC.

### Pathway mutual exclusivity

Oncogenic mutations in genes within the same oncogenic signaling pathway tend to not co-occur in the same patient due to functional redundancy[39,40]. In NSCLC, oncogenic mutations in the RTK/RAS, NRF1 and TP53 pathways have been shown to exhibit mutual exclusivity[40]. To demonstrate that reclassified pathogenic mutations have comparable cancer-driving effect on a pathway as known oncogenic mutations, we aimed to identify whether reclassified pathogenic mutations were mutually exclusive with other known oncogenic mutations in these three pathways within the MSK-IMPACT NSCLC data using two-sided Fisher's exact tests. All methods were able to identify VUSs that exhibit mutual exclusivity with other oncogenic mutations in all three pathways (Fig. S10A).

Within the NRF2 pathway, *KEAP1* VUSs reclassified as pathogenic by any method were mutually exclusive with oncogenic mutations in *KEAP1*, *NFE2L2* and *CUL3* (Fig. 2C), whereas *KEAP1* VUSs reclassified as benign by VEPs except for CADD, ESM1b, FATHMM and MutationAssessor tended to co-occur with other oncogenic mutations in the pathway (Fig. S10B). Similarly, all VEPs except for MutationAssessor were able to identify reclassified pathogenic mutations in *ATM* and *TP53* that are mutually exclusive with other oncogenic mutations in the TP53 pathway, whereas reclassified benign mutations in these genes were not mutually exclusive with other drivers in the pathway (Fig. 2C and S10).

In the RTK/RAS pathway, reclassified pathogenic mutations in *NF1*, *ERBB4*, and *IRS2*[41] identified by AlphaMissense were mutually exclusive with other oncogenic mutations (Fig. 2C). Reclassified

benign mutations in *ERBB4* and *IRS2* were also mutually exclusive with known oncogenic mutations (Fig. S10B), indicating potential drivers requiring additional annotation. *ERBB4* and *IRS2* pathogenic mutations classified by AlphaMissense were frequent in high TMB samples, while RTK/RAS oncogenic mutations were more common in TMB-low samples (Fig. S11A). Logistic regressions, controlling for TMB status, revealed independent mutual exclusivity between RTK/RAS oncogenic mutations and reclassified pathogenic mutations in *ERBB4* (Fig. S11B). A negative association was observed between *IRS2* pathogenic mutations and RTK/RAS oncogenic mutations, although not statistically significant due to limited sample size (Fig. S11C). Similar patterns were seen with reclassified benign mutations, suggesting potential unannotated drivers in *ERBB4* and *IRS2* (Fig. S11C). More details about the mutational pattern of RTK/RAS pathway and TMB status of samples with *ERBB4* and *IRS2* mutations are summarized in Supplementary Data 1.

We calculated the odds ratio of reclassified oncogenic mutations, compared to reclassified benign mutations, being mutually exclusive with other known oncogenic mutations in the same pathway ($OR_{mutex}$), although this analysis was largely underpowered at the gene level (Fig. S14D). The majority of these genes are oncogenes commonly altered with gain-of-function mutations, whose effects are more difficult to predict (Fig. 1B). In summary, analysis of mutational patterns showed that reclassified pathogenic mutations followed expected patterns within oncogenic pathways, offering a potential benchmark for VEP performance, even though most methods could improve their sensitivity to fully explore less common driver classes.

## Discussion

Results from our benchmarks suggest that there is not one single method that outperforms others across all tasks, although Alpha-Missense, SIFT and PolyPhen2 demonstrate better performance than other methods in multiple tasks related to reclassifying VUSs according to our proposed metrics (Fig. S14). Users should compare performance of multiple methods across specific evaluation tasks and choose the best performing methods for the tasks they are most interested in. Various tools exist to facilitate the comparison of different methods, including bioinformatics platforms, such as Ensembl VEP[42], dbNSFP[43] and OpenCRAVAT[44], that enable simultaneous annotation of variants with many VEPs, as well as cancer data portal such as cBioPortal[45], which integrates AlphaMissense predictions along with other cancer-specific variant annotations. Our preliminary experiments suggest that ensemble approaches that combine multiple VEP outputs can further improve the precision and robustness of pathogenicity predictions, providing a more comprehensive tool for clinical research and decision-making.

Our study has limitations. SNPs may have different distributions and characteristics compared to passenger mutations; thus, using SNPs as the negative set may overestimate VEPs performance in predicting mutation oncogenicity in tumors. Indeed, when we used neutral somatic variants as annotated by OncoKB instead of SNPs as the negative class, prediction performance declined across all methods (Figure S17). This result, however, also highlights the challenges with annotating true benign mutations in cancer, as annotations from knowledge bases often rely on evidence from functional experiments, which can be scarce or incomplete depending on the experimental setup. We expect that beyond reclassifying VUSs, the approaches described here can be used to review and revise existing variant annotations, including those neutral variants. Our VUS quantification across institutions comes from multiple contributing cancers with their own sequencing pipelines, the majority of which are tumor-only sequencing. Even though all data went through germline SNP filtering pipeline before public release, it is possible that there remained private SNPs in the data, which may artificially inflate the number of more easily characterizable VUSs in a given dataset, although this would

reflect a clinical reality of such SNPs appearing in tumor-only sequencing assays. The MSK-IMPACT and GENIE BPC cohorts, though richly annotated, may not be sufficiently powered at the present to discover rare driver variants in less commonly mutated genes or assert the association between these putative drivers and outcomes.

Overall, our findings underscore the potential of VEPs in identifying driver mutations in cancer as evidenced through their success at several benchmarks based on real-world data. VEPs can help to quickly expand the set of potential driver variants in genes with potential therapeutic significance such as *KEAP1*[46,47] and *SMARCA4*[48,49], which are targets of therapies currently undergoing clinical trials. We expect that VEPs will continue to improve over time, particularly with regard to cancer driver prediction; that real-world datasets fueling these analyses will continue to grow; and that a growing number of molecularly targeted therapies will allow for examination of not prognostic but also predictive value for identified genomic targets, together suggesting synergistic means by which data and computation can improve the lives of patients with cancer.

## Methods

This study complies with all relevant ethical regulations as approved by the Institutional Review Board of MSKCC.

### Patients and data collection

This study analyzed patients with tumor genomic sequencing from two sources: The MSK-IMPACT cohort and the AACR Project GENIE cohort, which includes patients from the MSK-IMPACT cohort.

**MSK-IMPACT.** The MSK-IMPACT cohort comprised patients at Memorial Sloan Kettering (New York, NY), an academic cancer hospital with tumor genomic sequencing using MSK-IMPACT, an FDA-authorized tumor genomic profiling assay, which uses matched white blood cell sequencing to filter clonal hematopoietic and germline variants. All MSK patients were enrolled as part of a prospective sequencing protocol (NCT01775072). The study was independently approved by the Institutional Review Board of each site. Patients provided written, informed consent and were enrolled in a continuous, nonrandom fashion. Data here is from a February 15, 2023 snapshot, consisting of 11,649 samples from 7965 patients with non-small cell lung cancer (NSCLC). Out of 7965 patients, 338 patients (4.2%) self-identified as Hispanic.

For patients in the MSK-IMPACT cohort, demographic and clinical information, including tumor stage, age, sex, race and histology were retrieved from the electronic health records database. Breakdown of demographic characteristics is presented in Table S5. Sex (female/male) was available and was included as a covariate in some models, but the primary analyses were not stratified by sex, as sex was not a primary variable of interest. Smoking history, prior treatment and metastatic events were abstracted from clinical notes using previously validated natural language processing methods[50–53]. Tumor mutation burden per sample was calculated as the total number of nonsynonymous mutations divided by the actual number of bases analyzed, and samples with TMB > = 10 mut/Mb were defined as TMB-high. MSI-H status was defined for each sample by an MSIsensor score >10[54].

**GENIE.** Details of the AACR Project GENIE cohort have been published previously[4]. In short, the pan-cancer registry contains genomic and clinical data from 11 international institutions. In this study, we analyzed data from the v.14-public release, which consists of genomic data for 183,302 tumors from 160,965 patients. For genomic landscape analyses involving gene-level counts, only patients with tumor sequencing panels including a given gene of interest were included in the respective analysis. The number of patients with each gene sequenced is given in Supplementary Data 2.

**GENIE BPC.** Genomic and clinical data for a subset of patients with non-small cell lung cancer in the AACR GENIE cohort have been recently published as part of the AACR GENIE Biopharma Collaborative (BPC)[37]. Out of 1846 patients from four contributing institutions in the cohort, we included all 977 patients from Dana-Farber Cancer Institute, Vanderbilt-Ingram Cancer Center and Princess Margaret Cancer Centre-University Health Network with single-primary NSCLC in our analysis cohort. All MSK patients were included in the MSK-IMPACT cohort. For survival analyses involving gene-level cohorts, only patients with tumor sequencing panels including a given gene of interest were included in the respective analysis.

### Genomic landscape

Genomic data, including mutational calls, copy number alteration and structural variant data, for the GENIE v.14-public cohort were obtained from Synapse. All genomic alterations were annotated with OncoKB version 4.2 (release date February 10, 2023). Genes were labeled as oncogenes or tumor suppressor genes using the OncoKB Cancer Gene List (updated October 2, 2023). For genomic landscape analyses involving gene-level counts, only patients with tumor sequencing panels including a given gene of interest were included in the respective analysis.

### Non-oncogenic variants

Non-oncogenic missense mutations were randomly selected from the dbSNP Human Variation Sets build 150 (April 2017) labeled as having "no known medical impact." This dataset includes variants with germline minor allele frequency of ≥ 0.01 and no records of clinical phenotypes in ClinVar. We annotated all variants with the same methods as described below and selected 7474 variants with the highest number of available annotations from 14 VEPs. These non-oncogenic variants served as the negative class in subsequent receiver operating characteristic curve analyses.

### Predicting established pathogenic variants

We evaluated the performance of 14 variant function prediction methods and one human variant archive for recapitulating known oncogenic cancer variants as annotated by OncoKB, the first FDA-recognized somatic molecular knowledge database for this purpose. Methods were chosen to be included from a diverse range of approaches based on their recent development and conceptual advancements (including methods with recent release dates e.g. AlphaMissense, PrimateAI and ESM1b, as well as more updated methodology to an old approach, e.g. EVE's deep generative model to predict pathogenicity based on evolutionary conservation), superior performance compared to other methods in the same class (e.g. REVEL and CADD outperformed other VEPs in multiple comparison studies[23,24], while VARITY_R_LOO's performance was better than 12 other VEPs and comparable to AlphaMissense in certain evaluation tasks[9]), specific relevance to cancer (FATHMM implements cancer-specific pathogenicity weight[55], MutationAssessor previously demonstrated utility in annotating cancer variants[12], CHASMplus[29] and BoostDM[30]'s cancer type-specific model of variant pathogenicity), as well as historical significance and popularity (SIFT and PolyPhen2 are among the earliest VEPs and have the highest number of citations to date[56]). A brief description of the methods/database is presented in Table S1.

### Annotation schema

OncoKB annotations were performed with version 4.2 (released February 10, 2023). Prediction scores for most VEPs, except BoostDM and CHASMplus and ClinVar were obtained from dbNSFP[43] v4.6 (released February 18, 2024). Pre-computed BoostDM scores were obtained from https://www.intogen.org/boostdm/search. CHASMplus

annotations were obtained from OpenCRAVAT using the pan-cancer model for GENIE v14 data and lung adenocarcinoma model for MSK-IMPACT NSCLC and GENIE BPC NSCLC data. Categorization of variants into pathogenic, non-pathogenic or uncertain, obtained by imposing predetermined cutoff thresholds on the predicted functional scores, was also provided by AlphaMissense, BoostDM, FATHMM, MutationAssessor, PolyPhen2 and SIFT. For VEPs in this category, we used off-the-self classifications provided by these methods. CHASMplus provided p-values for statistical significance of the predicted pathogenicity compared to a background model[29], so we corrected the p-values for multiple hypothesis testing using the Benjamini-Hochberg procedure and annotated variants with $q$-value $\leq 0.05$ as pathogenic and the rest non-pathogenic. The rest of the evaluated VEPs required additional steps to determine the appropriate variant classification from predicted scores.

First, EVE provides variant classifications at different degrees of uncertainty, aiming to maximize accuracy by excluding variants that the model is uncertain about. For example, a Class25 EVE classification means that 25% of the most uncertain variants were excluded when making predictions[28]. To identify the degree of uncertainty that would maximize accuracy in our data, we compared the AUROCs for classifying known oncogenic mutations from benign dbSNPs and selected the uncertainty threshold that resulted in the highest AUROC. Finally, for VEPs that provided prediction scores but not off-the-shelf classification or recommended score cutoffs for classifications, including CADD, ESM1b, PrimateAI, REVEL and VARITY, we manually identified cutoff thresholds to classify pathogenic and non-pathogenic mutations. To identify cutoff, for each method, we set up a binary classification task in which known oncogenic mutations from the dataset represented the positive class, and the benign dbSNPs represented the negative class. We then calculated the sum of sensitivity and specificity of each method at different thresholds and identified the optimal cutpoint as the threshold where this sum was maximized. In cases where more than one cutpoint were found, we used their median as the final threshold.

After cutoffs were determined, we applied them to the VEPs' predicted scores to stratify variants into categories. For CADD, PrimateAI, REVEL and VARITY_R_LOO, variants with scores higher than or equal to the determined thresholds were classified as pathogenic and vice versa. For ESM1b, variants with scores smaller than or equal to the determined thresholds were classified as pathogenic.

We repeated this process for all datasets, resulting in data-specific cutoff and EVE uncertainty thresholds. The cutoffs used are summarized in Table S2.

## Receiver operating curves (ROCs)

We performed two ROC analyses, 1. Weighting all positive variants equally and 2. To better understand method performance in a manner that reflects actual population levels of a given mutation, sampling positive variants from the GENIE cohort, i.e. appearing proportionally to their frequency in a real-world population. In the first analysis, mutation-level ROC for each method was constructed using all 8033 OncoKB-oncogenic missense mutations as the positive class and 7474 dbSNPs as the negative class. In the second analysis, population-level ROCs were constructed using all occurrences of OncoKB-oncogenic missense mutations in the GENIE v14-public cohort as the positive class ($N = 180{,}540$) and dbSNPs ($N = 7474$) as the negative class. The non-oncogenic mutations were upsampled to match the number of oncogenic mutations in the positive class, resulting in a balanced $N = 180{,}540$ for each class. Area under the curve and 95% confidence interval were calculated for each curve.

## Ligand binding and protein-protein interaction residues analysis

Residues involved in binding ligands, including small molecules, peptides, DNA and RNA, were retrieved from BioLiP2[57], a curated database of biologically relevant protein-ligand interactions. Residues important in protein-protein interactions (PPI), termed PPI hotspots and defined as residues whose replacements decrease the binding free energy significantly, were retrieved from PPI-HotspotDB[58]. We grouped missense mutations in either the GENIE v14-public or MSK-IMPACT NSCLC cohort into binding residues (including ligand-binding residues and PPI hotspots) versus non-binding residues. Fisher's exact tests were performed to test the enrichment of mutations occurring at binding residues for being reclassified as pathogenic by different methods.

## Pathway analysis

**Mutual exclusivity test.** Pathway analyses were done on the ten canonical oncogenic signaling pathways, including cell cycle, Hippo, Myc, Notch, Nrf2, PI-3-Kinase/Akt, RTK-RAS, TGFβ signaling, p53 and β-catenin/Wnt[40]. The gene lists constituting each pathway were retrieved from https://www.cell.com/cms/10.1016/j.cell.2018.03.035/attachment/73b4efd7-1e36-4e1f-874b-db6bc0a18ec4/mmc3.xlsx[40].

For each gene within a pathway, we aimed to identify whether reclassified pathogenic mutations in that gene are mutually exclusive with known oncogenic mutations in all genes within the same pathway. The result of this analysis demonstrates that reclassified pathogenic mutations have comparably pathogenic effect on a pathway as known oncogenic mutations. To this end, we calculated one versus all mutual exclusivity for each gene for patients with NSCLC in the MSK-IMPACT cohort. For each test, we first set up a 2 × 2 contingency table with two variables: the number of patients carrying reclassified pathogenic mutations in that gene, and the number of patients carrying oncogenic mutations in all genes within the same pathway. A two-sided Fisher's exact test was applied to the contingency table to test for mutual exclusivity.

An example contingency table used to test for pathway mutual exclusivity between *KEAP1* and all genes in the NRF2 pathway, including *KEAP1*, *CUL3* and *NFE2L2*, is below:

| | | Mutations in KEAP1 | |
| --- | --- | --- | --- |
| | | Reclassified pathogenic | VUSs/no mutation |
| Mutations in KEAP1, CUL3 and NFE2L2 | Known oncogenic | | |
| | VUSs/no mutation | | |

In particular, this table is used in a Fisher's exact test for mutual exclusivity between *KEAP1* and all genes in the NRF2 pathway. Two other tests were set up to test for mutual exclusivity of *NFE2L2* and *CUL3* with oncogenic mutations in NRF2 pathway genes.

The procedure was repeated for all genes present in a given pathway, and for all eight methods. P-values were adjusted for multiple hypothesis testing using the Benjamini-Hochberg procedure. Tests with a logOR < 0 and adjusted $p$-value <= 0.1 were considered significant for mutual exclusivity. The pathway-level one versus all mutual exclusivity rate was then calculated for each pathway by dividing the total number of significant mutually exclusive tests by the number of genes in the pathway.

As negative controls, we tested for mutual exclusivity between reclassified benign mutations and known oncogenic mutations in all genes in a given pathway using the same procedure.

**The role of tumor mutational burden in observed mutual exclusivity.** To identify whether *ERBB4* and *IRS2* AlphaMissense reclassified pathogenic mutations are mutually exclusive with RTK/RAS oncogenic mutations independent of TMB-high status, we performed a logistic

regression (Eq. 1):

$$RTK/RAS\ oncogenic\ mutations \sim AlphaMissense\ mutations + TMB - Hstatus \tag{1}$$

Where

*RTK/RAS oncogenic mutations* is 1 if a sample has any oncogenic mutations in RTK/RAS pathway genes, 0 otherwise

*AlphaMissense mutations* is 1 if a sample has an AlphaMissense reclassified pathogenic mutation in a gene of interest (*ERBB4* or *IRS2*), 0 otherwise

*TMB-H* is 1 if a sample has TMB >= 10 mut/Mb, 0 otherwise

Two independent regressions were run for *ERBB4* and *IRS2*.

## Survival analysis

To test the association of gene-level pathogenicity annotations with overall survival we performed a series of Cox proportional hazards (PH) models from time of diagnosis to time of death or last follow-up, left truncated at time of cohort entry (tissue sequencing). For patients with multiple sequencing events, the first was used as the time of cohort entry. To adjust for confounding variables between comparison groups, inverse probability of treatment weights (IPTW) were calculated using covariate values at baseline, including tumor stage, age, sex, race, histology, smoking history, tumor mutational burden (TMB), microsatellite instability status (MSI), prior treatment and metastatic sites if any, before fitting Cox PH models. An example plot of standardized mean differences in covariates before and after IPTW matching to demonstrate how IPTW helps achieve balance in covariates between comparison groups is presented in Fig. S5.

Hazard ratios, 95% confidence intervals and *p*-values for gene-level associations between a "pathogenic" alteration vs no alteration were computed. In the MSK-IMPACT cohort, all genes altered in >2% of the cohort were considered. For each gene of interest, only patients with tumor sequencing panels including a given gene in the target region were included in the analysis. The following gene mutation annotation schema was used: For OncoKB, any alterations annotated as oncogenic or likely oncogenic were used. For all other databases, any pathogenic alteration considered a variant of unknown significance in OncoKB (i.e. a "reclassified" pathogenic alteration) was used. q-values were computed using the Benjamini-Hochberg method; a false discovery rate 0.1 across all comparisons described in this analysis was used to determine statistical significance according to a prespecified statistical analysis plan (see 12-245 Appendix C Project Plan). The BPC cohort was used as a confirmation dataset in which only significant associations from the MSK-IMPACT analysis were tested; q-values were computed similarly but for only the number of hypotheses tested in the BPC.

**Kaplan-Meier curves.** Weighted Kaplan-Meier (KM) curves were constructed to further examine the relationship between gene-level pathogenicity annotations and overall survival for genes with significant hazard ratios in univariate weighted Cox PH models. Similar to the Cox PH regressions, KM curves were calculated from time of diagnosis to time of death or last follow-up, left truncated at time of cohort entry (tissue sequencing). Patients were stratified based on the presence of OncoKB oncogenic, 'reclassified' pathogenic, 'reclassified' benign mutations or without any mutation in each gene of interest. Only strata with ≥ 10 patients were included. KM curves were weighted using the same IPTWs used for the corresponding Cox PH regression.

**STK11/KEAP1 concurrent mutation analysis.** To test whether Alpha-Missense reclassified pathogenic variants in *STK11* and *KEAP1* were similarly associated with worse survival, we compared overall survival of patients with double *KEAP1* and *STK11* reclassified mutations with patients with single reclassified mutation and patients without any

mutation in these two genes. Patients with reclassified pathogenic mutations in both *KEAP1* and *STK11* are labeled as *KEAP1/STK11* double mutant, while patients carrying only reclassified pathogenic mutations in either gene are labeled as single mutant for the respective genes. Patients carrying oncogenic mutations in either gene are excluded from this analysis. Weighted KM curves were constructed as described above to compare overall survival of patients in these four strata.

## Random forest ensemble

Random forest ensembles were fitted using the R *caret* package to predict variant pathogenicity. The models incorporated scores from 11 non-ensemble methods evaluated in this study. OncoKB oncogenic variants served as the positive class, while variants randomly sampled from dbSNP were used as the negative class. The dataset was split into a 75:25 ratio for training and testing to ensure an unbiased assessment of model performance. Additionally, we trained separate RFs on mutation level data, where each unique mutation is only counted once, and on population level data, where all occurrences of mutations are included to reflect population distribution.

Before training, scores from each method were normalized and standardized to ensure consistency and comparability across the different methods. The 'preProcess' function in 'caret' was employed for this purpose, applying z-score normalization to center and scale the data.

Two cross-validation strategies were employed for model training: 1. randomly sampled 5-fold cross-validation and 2. gene holdout cross-validation. In the 5-fold cross-validation approach, the training set was randomly partitioned into five subsets. The model was trained on four subsets and validated on the remaining subset, rotating this process five times to ensure each subset served as the validation set once. In gene holdout cross-validation, variants were grouped by gene, and entire genes were withheld during training to serve as a validation set, assessing the model's ability to generalize to unseen genes.

Model performance was evaluated using metrics including accuracy, sensitivity, specificity, and area under the receiver operating characteristic curve (AUROC). Additionally, feature importance was assessed to determine the contribution of each method's scores to the overall model performance. The hyperparameter *mtry*, which represents the number of predictors that will be randomly sampled at each split when creating the tree models, was tuned to maximize AUROC, with mtry = 2 giving the best performance and thus selected for the final model.

All statistical analyses were performed in R version 4.2.2 (2022-10-21).

## Reporting summary

Further information on research design is available in the Nature Portfolio Reporting Summary linked to this article.

# Data availability

Source data are provided with this paper. The processed AACR Project GENIE v14 data is available on Synapse at https://www.synapse.org/Synapse:syn7222066. GENIE BPC NSCLC genomic and clinical data are available on Synapse at https://www.synapse.org/Synapse:syn27056172/wiki/616601. The MSK-IMPACT NSCLC genomic data is available on cBioPortal as part of the MSK-CHORD cohort at https://www.cbioportal.org/study/summary?id=msk_chord_2024. All clinical annotations used for analyses in this paper are available at https://github.com/clinical-data-mining/variant-annotation to enable others to reproduce our findings and make additional discoveries. The raw sequencing data for the AACR Project GENIE and MSK-IMPACT cohorts are protected and are not broadly available due to privacy restrictions and must be requested with appropriate institutional approvals. Source data are provided with this paper.

## Code availability

R codes used in the analysis of this paper are available under MIT license on GitHub at https://github.com/clinical-data-mining/variant-annotation. Additionally, a code container is available on Zenodo at https://doi.org/10.5281/zenodo.15421055[59].

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

## Acknowledgements

We thank members of the Schultz Lab and the MSK Cancer Data Science Initiative for helpful discussions and feedback on this work. This work was supported by the MSK Support Grant/Core Grant (P30 CA008748), the MSK Molecular Diagnostics Service in the Department of Pathology, the Marie-Josee and Henry R. Kravis Center for Molecular Oncology, the Halvorsen Center for Computational Oncology, and K08CA286842 (to J.J.).

## Author contributions

T.N.T., J.J., and N.S. conceptualized the research aims and devised the experiments. T.N.T. curated the data and performed data analyses. C.F., K.P., A.L., M.W., S.K., X.L., and I.B. provided data and/or analysis and software pipelines that enabled the data analyses in this study. R.S. and Y.C. provided biostatistic input and supervision. T.N.T., M.F.B., G.R., M.L., D.C., N.S., and J.J. contributed to the interpretation of the results. N.S. and J.J. supervised the project. T.N.T., J.J., and N.S. wrote the manuscript. All authors provided critical feedback and helped shape the research, analysis, and manuscript.

## Competing interests

The authors declare the following competing interests: G.R. declares professional services and activities (uncompensated) for the American Association for Cancer Research, the American Society of Clinical Oncology, Mirati Therapeutics, Pfizer, Takeda Pharmaceuticals and Verastem; and professional services and activities for Harborside Press, MJH Associates, the National Comprehensive Cancer Network, Phillips Gilmore Oncology Communications, Research to Practice and Triptych Health Partners. N.S. declares professional services and activities (uncompensated) for Cambridge Innovation Institute and Innovation in Cancer Informatics; and professional services and activities for Novartis and OneOncology. M.L. declares equity in and professional services and activities (uncompensated) for Paige.AI. M.F.B. declares professional services and activities for AstraZeneca and Paige.AI; professional services and activities (uncompensated) for JCO Precision Oncology and the Journal of Molecular Diagnostics; and intellectual property rights in SOPHiA GENETICS. J.J. has a patent licensed by MDSeq. T.N.T., C.F., K.P., A.L., M.W., S.K., X.L., I.B., R.S., Y.C., and D.C. declare no competing financial and non-financial interests.
