## [Peer Review File · Nature Communications]

AI cancer driver mutation predictions are valid in real-world data

Corresponding Author: Dr Nikolaus Schultz

Version 1:

Reviewer comments:

Reviewer #2

(Remarks to the Author)

The authors have addressed all points I previously raised. The article now provides a systematic and thorough evaluation of VEPs in a variety of different scenarios. I believe it will be a valuable resource for scientists looking to predict somatic variant effects in cancer.

I only have a few minor comments:

- * Results / "Association with OncoKB driver variants" / final paragraph / second line: "Tot est" => "To test"
- * Results / "Ensemble predictors of variant effects": Subsection header with no text, remove?

(Remarks on code availability)

Reviewer #3

(Remarks to the Author)

I appreciate the effort done by the authors to take into account and incorporate my comments and those of the two other reviewers in the new version of the manuscript.

As anticipated (which can be clearly appreciated in the new version of Figure 1 and Figure S16), the distribution of scores of variants that follow the mutational profile of tumors is different from that of SNPs and more closely resembles the distribution of VUS. I think that given this fact, the AUROC presented by the authors should be constructed with neutral somatic variants as negative set, rather than with SNPs, as this will more closely track the performance of the VEPs in separating neutral from oncogenic somatic variants.

(Remarks on code availability)

Version 2:

Reviewer comments:

Reviewer #3

(Remarks to the Author)

The authors have addressed the concern raised in the previous round of reviews. I have no further comments.

(Remarks on code availability)

Reviewer #1

Remarks to the Author

EDITORIAL NOTE: This Reviewer was not able to submit their report. However, Reviewer #3 assessed your response and considered that Reviewer #1's concerns were addressed.

Reviewer #2

Remarks to the Author

The authors have addressed all points I previously raised. The article now provides a systematic and thorough evaluation of VEPs in a variety of different scenarios. I believe it will be a valuable resource for scientists looking to predict somatic variant effects in cancer.

I only have a few minor comments:

- * Results / "Association with OncoKB driver variants" / final paragraph / second line: "Tot est" => "To test"
- * Results / "Ensemble predictors of variant effects": Subsection header with no text, remove?

Response

Thank you. We have corrected the typographical and formatting errors.

Reviewer #3

Remarks to the Author

I appreciate the effort done by the authors to take into account and incorporate my comments and those of the two other reviewers in the new version of the manuscript.

As anticipated (which can be clearly appreciated in the new version of Figure 1 and Figure S16), the distribution of scores of variants that follow the mutational profile of tumors is different from that of SNPs and more closely resembles the distribution of VUS. I think that given this fact, the AUROC presented by the authors should be constructed with neutral somatic variants as negative set, rather than with SNPs, as this will more closely track the performance of the VEPs in separating neutral from oncogenic somatic variants.

Response

We thank the reviewer for this suggestion. We have conducted additional analyses where AUROCs were calculated using known oncogenic mutations as the positive class and known neutral somatic mutations (experimentally confirmed and catalogued in OncoKB) as the negative class (**Figures S17**). We included genes with at least 10 neutral and 10 oncogenic mutations in these analyses and calculated AUROCs at both population level, where each occurrence of a mutation was included, and at mutation level, where each unique mutation

was included once. Even though AUROCs from all methods decreased due to the smaller number of annotated neutral somatic mutations, we found that the overall trends we observed in **Figure 1B** held: overall, VEPs based on structure prediction/protein language model achieved the highest performance, with AlphaMissense achieved the highest AUROCs in predicting both mutations in oncogenes (OGs) and tumor suppressor genes (TSGs) (AUROC = 0.80 and 0.96 respectively at population level). Additionally, all methods performed better in predicting oncogenic variants in TSGs compared to those in OGs.

We acknowledge the reviewer's concern that SNPs do not fully capture the distribution of passenger mutations in tumors, as reflected in the score distributions in **Figure 1C** and **Figure S3**. However, accurately annotating true neutral somatic variants remains challenging. Knowledge bases such as OncoKB rely on evidence from functional experiments, yet a mutation that appears inconsequential in one cell line may have a damaging effect in another tumor type or treatment context. Similarly, a variant could have no effect on one experimental readout e.g. activity of a specific pathway, while disrupting others that remain unmeasured. Additionally, because databases like OncoKB rely on published studies, the limited number of curated neutral variants is often concentrated in well-studied genes like *TP53* and *BRCA2*, introducing potential bias. While SNPs may serve as an imperfect negative control, the performance of VEPs reported in the main paper is supported by our additional analyses, where AUROCs were calculated using known oncogenic mutations as the positive class and known neutral somatic mutations as the negative class (**Figure S17**). Given these challenges, we propose that the approaches outlined in our manuscript could help refine current variant annotations and identify neutral mutations with possible oncogenic effects.

We have added the following text in the main article to discuss this point:

*“Additionally, to more accurately reflect the type and distribution of passenger mutations in tumors, we used neutral somatic mutations as annotated by OncoKB as the negative class to evaluate VEP performance (**Figure S17**). We found that performance declined across methods, even though deep learning and ensemble methods tended to perform better than evolution-based methods (**Figure S17**). Out of all methods, AlphaMissense achieved the best performance in this task (AUROC of 0.8 for classifying OGs and TSGs mutations at the population level respectively, **Figure S17B**).*

*SNPs may have different distributions and characteristics compared to passenger mutations, thus, using SNPs as the negative set may overestimate VEPs performance in predicting mutation oncogenicity in tumors. Indeed, when we used neutral somatic variants as annotated by OncoKB instead of SNPs as the negative class, prediction performance declined across all methods (**Figure S17**). This result, however, also highlights the challenges with annotating true benign mutations in cancer, as annotations from knowledge bases often rely on evidence from functional experiments, which can be scarce or incomplete depending on the experimental setup. We expect that beyond reclassifying VUSs, the approaches described here can be used to review and revise existing variant annotations, including those neutral variants.”*